# mHealth guideline training for non-communicable diseases in primary care facilities in Nigeria: a mixed methods pilot study

Akaninyene Asuquo Otu [1,2] Emmanuel E Effa,[1,2] Obiageli Onwusaka,[2,3] Chiamaka Omoyele,[4] Stella Arakelyan [5] Okey Okuzu,[6] John Walley[7]

For numbered affiliations see end of article.

**Correspondence to**
Dr Akaninyene Asuquo Otu;
akanotu@yahoo.com

## ABSTRACT

**Objective** To pilot the use of a scalable innovative mobile health (mHealth) non-communicable diseases (NCDs) training application for nurses at the primary care level.

**Design** Mixed methods pilot of mHealth training on NCD care for nurses at primary healthcare (PHC) facilities. We provide a descriptive analysis of mHealth training test scores, with trend analysis of blood pressure (BP) control using paired t-test for quantitative data and thematic analysis for qualitative data.

**Setting** PHC facilities in rural and urban communities in Cross River State, south eastern Nigeria. NCDs were not part of routine training previously. As in most low-and-middle-income settings, funding for scale-up using conventional classroom in-service training for NCDs is not available in Nigeria, and onsite supervision poses challenges.

**Participants** Twenty-four health workers in 19 PHC facilities.

**Intervention** A self-paced mHealth training module on an NCD desk guide was adapted to be applicable within the Nigerian context in collaboration with the Federal Ministry of Health. The training which focused on hypertension, diabetes and sickle cell disease was delivered via Android tablet devices, supplemented by quarterly onsite supervision and group support via WhatsApp. The training was evaluated with pre/post-course tests, structured observations and focus group discussions. This was an implementation pilot assessing the feasibility and potential effectiveness of mHealth training on NCD in primary care delivery.

**Results** Nurses who received mHealth training recorded a statistically significant difference (p<0.001) in average pretest and post-test training scores of 65.2 (±12.2) and 86.5 (±7.9), respectively. Recordings on treatment cards indicated appropriate diagnosis and follow-up of patients with hypertension with significant improvements in systolic BP (t=5.09, p<0.001) and diastolic BP (t=5.07, p<0.001). The mHealth nurse training and WhatsApp support groups were perceived as valuable experiences and obviated the need for face-to-face training. Increased workload, non-availability of medications, facility-level conflicts and poor task shifting were identified challenges.

**Conclusions** This initiative provides evidence of the feasibility of implementing an NCD care package supported by mHealth training for health workers in

## STRENGTHS AND LIMITATIONS OF THIS STUDY

⇒ Pragmatic implementation of a mHealth training and support model based on a country-specific (WHO PEN compatible) non-communicable disease package of care.

⇒ Successful process evaluation of feasibility, implementation experience, challenges and patient satisfaction using quantitative and qualitative methods.

⇒ A decline in patient numbers attending facilities imposed by the COVID-19 pandemic and the shortened pilot period led to a reduction in numbers screened, diagnosed, started on treatment and evaluated.

⇒ Perception of the initiative predominantly as a nongovernmental organisation pilot rather than an obligatory health system strengthening initiative driven by the relevant authorities (State and Federal Ministries of Health).

PHCs and the strong possibility of successful scale-up nationally.

## INTRODUCTION

Non-communicable diseases (NCDs) are responsible for the deaths of 41 million people annually, with an estimated 15 million of these deaths occurring in people between 30 and 69 years of age.[1] Among NCDs, the four top conditions that together account for more than 80% of all these premature deaths include cardiovascular diseases (17.9 million deaths annually), cancers (9 million), respiratory diseases (3.9 million) and diabetes (1.6 million). Almost three-quarters of NCD-linked deaths occur in low-and-middle-income countries. Fragile health systems experience challenges in controlling both NCDs and infectious diseases such as HIV/AIDS and tuberculosis (TB). The rapid global epidemiological transition and rising rates of NCDs are also having negative effects on the health of Nigerians.

In Nigeria, NCDs are estimated to account for 29% of all deaths,[2] with 20% dying prematurely.[3] Community-based surveys involving adults in Cross River State Nigeria have identified that approximately 7% of residents in Calabar had undiagnosed diabetes mellitus,[4] 42% and 18% had hypertension and obesity, respectively.[5] It is projected that by 2030, NCDs will be the leading cause of morbidity and mortality in Nigeria.[6 7] This rise in NCDs has been attributed to rapid population growth and urbanisation, limited access to healthy and nutritious food and physical inactivity.[8 9]

The fragile Nigerian public health system is not well equipped to effectively manage and control NCDs.[10] Until 2019, NCD care was not prioritised by policy-makers. For example, only 3.3% of the national health budget is devoted to mental healthcare with fewer than 150 mental health specialists serving Nigeria's estimated 206 million population.[11] In 2019, Nigeria's Federal Ministry of Health launched the National Multisectoral Action Plan for the Control and Prevention of NCDs,[12] prioritising health system strengthening at all levels of care towards universal health coverage. A major thrust of the plan was the integration of NCD management into primary healthcare (PHC) services. Despite this, the recently launched National task-shifting and task-sharing policy does not cover NCD prevention and control.[13] NCDs are not prioritised in health worker basic or post-basic training, as in many other countries. So far, the PHC facilities in Nigeria have not been involved in programmatic/systematic care for NCDs (or other chronic diseases except for TB and HIV).

The Communicable Disease Health Service Delivery (COMDIS-HSD) programme has developed care guideline processes to identify, diagnose and treat NCDs. These packages of care have been adapted and used at a provincial scale in Pakistan, Swaziland[14] and elsewhere with higher rates of diagnosis and NCD control in the primary care facilities.[15–20] Since 2017, this NCD package has been scaled-up across the outpatient departments of all public hospitals and many health centres in the Punjab Province, resulting in the diagnosis and treatment of 540 000 people.[16–20] Similarly, the COMDIS-HSD NCD package has been adapted and adopted in Sierra Leone[21] and in 71 rural health facilities in China, covering 2.5 million people.[22]

The COMDIS-HSD NCD package has also been adopted by the Nigerian Federal Ministry of Health (FMoH) NCD technical working group, with technical support from the Nuffield Leeds, UK. The package is technically compatible with the WHO Package for Essential Noncommunicable (PEN) disease interventions and provides steps of medication and patient education (https://comdis-hsd.leeds.ac.uk/resources/ncd-care-package/). The PEN/COMDIS-HSD package has been piloted in its hard copy residential course format by the FMoH/NCD division in Abuja Federal Capital Territory. However, there is no funding for scaling-up these traditional in-class service courses for health workers at a national level.

By leveraging the high mobile phone penetration in Nigeria, we aimed to pilot the use of an innovative mobile health (mHealth) user-friendly video training (VTR) mobile application that can be used on any smart phone, tablet or computer both online and off-line. The guide and training modules were adapted for mHealth online delivery by Instrat Global Health Solutions (an indigenous technology company) with pre-post tests and award of continuing professional development certificates. We hypothesised that mHealth NCD training and a support group for nurses should lead to improved knowledge and improved competencies in the screening/diagnosis, treatment initiation and optimal blood pressure (BP) control in patients at the PHC level. Our overarching goal was to develop a workable and sustainable approach to support health workers in managing NCDs at the PHC level which can be scaled-up across Nigeria.

## METHODS
### Patient and public involvement
The development of the research question and the outcome measures in this initiative were directly informed by Nigerian patients' priorities, experience and preferences. These priorities were factored into the design of this study. Patients were not directly involved in the recruitment to and conduct of the study but were beneficiaries of the NCD package of care we provided in the PHCs. We aim to disseminate the results of this initiative by conducting interactive sessions with the patients who continue to receive care for their NCDs in the PHCs in Cross River State.

### Implementation package
The NCD desk guide, training modules and treatment card record package were adapted from the WHO PEN guidelines by our team (http://comdis-hsd.leeds.ac.uk) in collaboration with WHO Nigeria and the Federal Ministry of Health. In Nigeria, registered nurses, who are qualified health workers at PHC facilities, have not been trained to provide NCD prevention and treatment. The NCD package had been initially piloted in conventional face-to-face training at a venue in the Federal Capital Territory Abuja with WHO Nigeria funding. However, there was no funding for scaling-up this package across the states and thousands of PHC facilities. Online mHealth training and support have the potential for large-scale replication.

### Research design
In this pilot, we adopted a mixed methods approach with the following goals:
1. To evaluate changes in NCD-related knowledge through pre-post training assessments included with the online training.
2. To understand service delivery and implementation contextual factors through structured observations

**Table 1** Logic model table showing intervention inputs, intervention process and intended practice change

| Intervention inputs | Intervention process and actions | Intended Practice change | Outputs | Health outcome |
|---|---|---|---|---|
| ► Case management desk guideline, including lifestyle counselling<br>► mHealth training of nurses on the desk guide package<br>► Nurse WhatsApp support group<br>► Supervision of care visits (quarterly)<br>► Digital BP apparatus, glucometer and strips<br>► Treatment cards, used for clinical care (subsequently used also as research data records)<br>► Monthly funds for health workers' telecommunication fees | ► Screen/diagnose<br>► Prescribe antihypertensive<br>► Identify hypertension and/or diabetes and treat<br>► Counsel for lifestyle modification<br>► Follow-up care<br>► Focus group discussion with nurses, month 5 | Provider practice:<br>► Screen, diagnose, treat, counsel, follow-up and report as per the case management desk guideline<br>Patients practice:<br>► Follow-up visits<br>► Treatment<br>► Lifestyle changes as counselled, e.g., healthy eating, activity, smoking cessation | Patients are:<br>► Screened and diagnosed as per desk guide<br>► Prescribed right drug/ dose<br>► Counselled for lifestyle change<br>► Followed-up and treated, that is, continuing care | ► BP trend by visit (primary outcome)<br>► Hypertension and/ or diabetes care recorded on the NCD card |

BP, blood pressure; NCD, non-communicable disease.

and focus group discussions (FGDs) involving the health workers following training.

3. To assess outcomes through record review of treatment cards for appropriateness of screening, diagnosis and initiation of treatment and changes in BP, etc.

The implementation of this pilot was guided by the proposed theory of change that recognises the various inputs and activities expected to generate patient-related outcomes and health systems impact driven by the policy change. The theory is underpinned by contextual factors such as the current organisation, resourcing and limitations of the health system; limitations placed on the scope of tasks performed by health workers at PHCs; availability of funding to train health workers and the opportunity to leverage good mobile phone penetration. It was not possible to involve patients or the public in the design, conduct, reporting or dissemination plans of our research. Table 1 presents the logic model guiding the implementation of the pilot. The model captures the various inputs for this project including the case management and lifestyle counselling desk guidelines, the mHealth training, supervisory visits, treatment cards and relevant equipment. The model also reflects the actions undertaken in the course of this project, that is, screening for NCDs, prescription of relevant drugs, counselling on a healthy lifestyle, FGDs and follow-up care. These actions are then linked to intended practice changes, outputs and health outcomes.

## Setting and participants

This pilot was carried out in a total of 19 PHC facilities in Cross River State, South-Eastern Nigeria over 12 months between February 2020 and January 2021. It involved 23 nurses and one community health extension worker drawn from selected sites in rural areas and peri-urban areas in Cross River State. Purposive sampling was used to choose the 19 PHC facilities; this was done in collaboration with the Cross River State Primary Health Care Development Agency (CRSPHCDA). The CRSPHCDA is a government agency tasked with ensuring 70% of the population in CRS have access to affordable quality healthcare at the primary care level. The CRSPHCDA proposed 23 nurses and one community health extension worker from the 19 PHC facilities for the NCD training based on the number of available nurses per facility. On the average, there is one nurse per PHC facility. It was spread across all the senatorial zones and local government areas with the aim of sampling from at least 10% of the just over 190 larger PHCs (staffing including at least one registered nurse) in Cross River State.

## Process

A one-day face-to-face orientation was organised in Calabar, the capital of Cross River State, in February 2020. This introduced 24 health workers to the NCD training package, giving them hard copies of the desk guide, job aid, treatment cards and showed them how to access and use the mHealth training modules for self-study.

The mHealth modules were developed using the *MediXcel Lite* health technology platform which has been previously described.[23 24] The training modules consisting of video and text-based learning materials were created in English which was understood by all health workers. Several facilitation techniques were used by the trainers

such as practical sessions, group discussions and case studies. All health workers were then expected to complete the eight NCD modules over a 2-week period by accessing the NCD materials on their mobile phones and tablets. User logins were created for the health workers by the staff of InStrat to enable them to download the app onto their Android mobile phones. This app also tracked their progress with the NCD modules automatically. The maximum possible score was 100%, with the passing score set at 70% for each module. Continuing professional development certificates were awarded to health workers who completed the NCD modules.

Following this orientation and mHealth training, the health workers were expected to use the information contained in the modules to screen and identify patients presenting with NCDs to their various facilities. Treatment cards were distributed, and participants were instructed to upload monthly updates on the number of NCD cases seen in the various facilities via an electronic CommCare app portal on mobile electronic devices. Relevant clinical materials such as sphygmomanometers, glucometers and test strips were provided to facilitate the clinical assessment of clients with NCDs.

The health workers were provided with funds monthly to cover telecommunications fees. A community of practice WhatsApp group was created for the health workers to facilitate peer-to-peer interactions, real-time patient care support by clinicians within the research team and data management by IT personnel of the implementation team. The WhatsApp group also served as an avenue to provide clinical support and data recording support. The health workers who uploaded their facility data onto the CommCare app on time were openly commended as a form of non-financial motivation and as a reminder to others to do the same.

Supportive supervision was provided to the health workers in the form of weekly discussions, WhatsApp phone calls and supplemented by quarterly facility clinician supervisory visits. The facility visits involved first-hand assessments of the service delivery contextual factors. During these trips, clinical mentoring was provided, and the health workers were observed providing care to patients using the clinical desk guide and treatment cards. Feedback was obtained from the health workers during these sessions.

In response to the early implementation issues identified by the implementation team, a new module on integrating NCD management into the daily activities of PHC facilities was developed and deployed on the training app.

### Focus group discussions

In the fifth month from the start of implementation, FGDs were conducted as part of the research, with the health workers (see online supplemental file). Focus groups (FGs)[25] were the chosen method as they are useful for exploring perpectives and encouraging reflections on experiences. Short message service invitations were sent to 24 participants who completed the intervention to share their perceptions of the digital training and experiences in FGDs.[25]. Twenty-three participants accepted the invitation. We conducted four FG sessions over 10 days, with an average of six participants in each group (FG1 n=8, FG2 n=4, FG3 n=5, FG4 n=6). There were only two male participants, and they were part of the fourth group.

Each FGD was facilitated by two researchers experienced in conducting FGDs and an assistant. The research assistant coordinated the session schedules, recording and served as the official note-taker. Sessions were 1–2 hours in length and conducted face-to-face in locations that were convenient for participants. Open-ended questions, formulated by the team who have formal training in qualitative research, were used to gain the best perspective and scope of individual responses on (1) perceptions of the use of mHealth training strategy, (2) experiences with the implementation of clinical guidelines, (3) reflections on the adequacy of received supervisions, (4) reflections on potential scale-up of the intervention. Facilitator notes ensured uniformity among FGDs and kept each discussion on target. At each FGD, an opportunity was given to withdraw consent for participation and recording. Responses were recorded, and participants were assured their comments would be kept confidential and anonymous.

No formal sample size was calculated for this pilot study.

### Data analysis
#### Quantitative data
The screening, diagnosis, treatment and follow-up care are specified in the NCD guide, and this information is recorded on the treatment card. The record review was of all patients identified and initiated on care. The assessment was of quality care according to these guidelines and the completeness of recording of key clinical measurements of BP, body mass index (BMI), fasting blood sugar (FBS) and urine analysis (as per guideline for that disease) were metrics used to assess the effectiveness of the pilot project. We assessed the effect of the NCD mHealth training through the pre/post-test results and assessed operational issues that may have contributed to the results. The overall assessment was the degree to which the NCD mHealth training and care was feasible, sustainable, replicable and treatment initiation quality was (as per guideline). Phone interviews were conducted on a random sample of patients across all facilities using a brief guide that included questions on satisfaction.

Data were imported and analysed in SPSS V.23.[26] Descriptive analysis of quantitative data was carried out using means, SD, frequencies and percentages. Group differences were established using $\chi^2$ analysis. Descriptive trend analysis for the follow-up rates in the systolic BP and diastolic BP changes in patients with hypertension was conducted on the monthly basis. The paired t-test was used to compare the average change in systolic BP and diastolic BP levels at baseline and 3-month follow-up among patients with hypertension. Hypothesis testing was two-tailed, at the 5% level.

### Qualitative data

We used qualitative thematic analysis to identify prominent themes and patterns in the data and included a combination of deductive and emergent strategies. The facilitators debriefed immediately after each FGD. Two researchers (AAO and SA) independently reviewed the data and came together to discuss emerging ideas. After the discussion, the researchers returned to the data to further review and confirm coding. This iterative process resulted in grouping the codes into three major themes: (1) perceptions with the mHealth trainings, (2) experiences with the implementation of NCD care packages, (3) potential issues with the intervention scale-up. Qualitative data were analysed using Dedoose software.[27]

The development of the research question and the outcome measures in this initiative were directly informed by Nigerian patients' priorities, experience and preferences. These priorities were factored into the design of this study. Patients were not directly involved in the recruitment to and conduct of the study but were beneficiaries of the NCD package of care we provided in the PHCs.

## RESULTS

A total of 24 health workers from 19 PHCs across the state were trained using the app. All the participating health workers achieved the passing score of 70% for each module, enabling them to progress to the next module. There was no limit on the number of attempts that could be made on the post-module test, and all the health workers were able to complete the modules in the given 2 weeks. The average pre-test and post-test scores were 65 (±12) and 87 (±8), respectively, with a statistically significant difference between these scores following a two-tailed t-test ($p < 0.001$).

Since the project initiation, 414 (72.7%) community members of the 569 patients seen were newly diagnosed cases of NCDs. The rest were patients who had previously been diagnosed with NCDs prior to the onset of the project. The average age of males diagnosed with an NCD was 53 years (SD=15) and for females was 49 years (SD=14). Over three-quarters (n=432, 76%) of all cases and over half (n=315, 55%) of newly diagnosed cases were patients with hypertension (newly diagnosed:110 male, 205 female, average age 52, SD=13) and 14% were diabetics (15 male, 35 female, average age 49, SD=14) (table 2).

FBS checks were conducted on 215 of the community members, with 47 (22%) of them falling within the prediabetes range of 5.6–6.9 mmol/L and 82 (38%) falling above the diabetes threshold of >7.0 mmol/L (table 2). Proteinuria and/or glycosuria were identified in 14 of the 113 persons who had a urine analysis. In accordance with the NCD guideline, fasting blood glucose was only done if the person was overweight or had related symptoms, and urine analysis was only done as indicated.

Gender distribution of newly diagnosed NCD cases (figure 1) was not statistically different $\chi^2$ (6)=8.4, p=0.2. On average, 16 patients with hypertension and 3 patients with diabetes were identified per facility (range 3–38 for patients with hypertension and 0–12 for patients with diabetes).

All patients with NCDs (n=569) registered at targeted facilities received height (μ=1.56, SD=0.12) and weight measurements (μ=68.1, SD=12.47) at the first appointment. BMI was calculated and recorded for all of these patients (μ=28.1, SD=5.85). Over 40% of patients with NCDs (n=234, 41%) received an appointment reminder following the screening appointment.

### Healthy lifestyle advice

In all, 356 (62.6 %) patients were given lifestyle advice such as reducing/stopping cigarette smoking and alcohol, reducing weight, reducing salt intake, increasing physical activity/exercise, reducing stress, medication adherence, healthy eating and healthy living. For these, a motivational counselling strategy was used.

### Primary outcome

Following the initial screening visit involving 420 patients, the patients were followed up with 230 of them making two such follow-up visits. A total of 98 patients made up to three follow-up visits (table 3). Recordings on treatment cards indicated a significant improvement in BP of patients with hypertension from visit 1 through to visit 3 (figure 2). The systolic BP for patients with hypertension who had three follow-up visits (after the initial screening visit) steadily decreased from the visit 1 average of 162 to a visit 3 average of 139 mm Hg. The average systolic BP decreased 23 mm Hg between the screening to follow-up visit 3 (p<0.001). A similar trend was registered as a decrease of 12 mm Hg in diastolic BP from the follow-up visit 1 average of 93.8 mm Hg to follow-up visit 3 average of 81.7 mm Hg (p<0.001).

### Compliance with medication

Patients with hypertension were prescribed antihypertensive drugs from the following classes: (1) ACE inhibitors (lisinopril, captopril); (2) calcium channel blockers (nifedipine, amlodipine); (3) diuretics (furosemide, hydrochlorothiazide) which were all in line with recommendations provided in the NCD desk guide. Where hydrochlorothiazide was unavailable, normoretic (i.e., amiloride hydrochloride 5 mg and hydrochlorothiazide 50 mg) was prescribed. The top three most often prescribed antihypertensive drugs were normoretic (50%), nifedipine (44%) and amlodipine (33%). Patients with diabetes were predominantly prescribed metformin, glibenclamide or insulin injections.

### Patient satisfaction

Patient satisfaction with nurses and clinical interaction was assessed based on patient exit interviews with 29 randomly selected patients with NCDs (17 hypertensive, 5 diabetic, 6 comorbid, 1 asthmatic) within 2 weeks of their

**Table 2** Characteristics of patients with NCDs served by targeted PHC clinics, n=569

| Characteristics | Male (n=206/36%) | Female (n=363/64%) | Total (n=569) |
|---|---|---|---|
| Age (μ±SD) | 53.5±15.2 | 49.75±14.2 | 51.8±14.2 |
| **Patients with NCD who attended facilities** | | | |
| Hypertension | 155 (75.2%) | 277 (76.3%) | 432 (75.9%) |
| Diabetes | 23 (11.2%) | 47 (13.0%) | 70 (12.3%) |
| Asthma/COPD | 10 (4.9%) | 17 (4.7%) | 27 (4.7%) |
| Depression | 1 (0.5%) | 8 (2.2%) | 9 (1.6%) |
| Sickle cell anaemia | 6 (3.0%) | 1 (0.3%) | 7 (1.2%) |
| Epilepsy | 3 (1.5%) | 3 (0.8%) | 6 (1.1%) |
| Other NCDs | 8 (3.9%) | 10 (2.8%) | 18 (3.2%) |
| Total | 206 (100%) | 363 (100%) | 569 (100%) |
| **New diagnosis of NCDs** | | | |
| Yes | 145 (70.4%) | 269 (74.2%) | 414 (72.7%) |
| No | 61 (29.6%) | 94 (25.8%) | 155 (27.3%) |
| Total | 206 (100%) | 363 (100%) | 569 (100%) |
| **Check-up items** | | | |
| Height (μ±SD) | 1.56±0.14 | 1.56±0.11 | 1.56±0.12 |
| Weight (μ±SD) | 67.4±12.6 | 68.5±12.4 | 68.1±12.5 |
| **BMI** | | | |
| Normal 18.5–24 | 57 (27.7%) | 114 (31.4%) | 171 (30.0%) |
| Overweight 25–30 | 78 (37.9%) | 124 (34.1%) | 202 (35.5%) |
| Obese ≥30 | 62 (30.1%) | 122 (33.6%) | 184 (32.3%) |
| Total | 206 (100%) | 363 (100%) | 569 (100%) |
| **BP** | | | |
| Normal BP 90/60–120/80 mm Hg | 52 (25.2%) | 73 (20.1%) | 125 (22.7%) |
| High normal 130/85–139/89 mm Hg | 21 (10.2%) | 38 (10.5%) | 59 (10.4%) |
| High BP ≥140/90 mm Hg | 133 (64.6%) | 252 (69.4%) | 385 (67.7) |
| Total | 206 (100%) | 363 (100%) | 569 (100%) |
| **Diagnostic tests** | | | |
| **FBS** | | | |
| Normal reading <5.6 mmol/L | 30 (40.0%) | 56 (40.0%) | 86 (40.0%) |
| Pre-diabetes 5.6–6.9 mmol/L | 15 (20.0%) | 32 (23%) | 47 (21.9%) |
| Diabetes reading >7.0 mmol/L | 30 (40.0%) | 52 (37.1%) | 82 (38.1%) |
| Total | 75 (100%) | 140 (100%) | 215 (100%) |
| **Urine analysis** | | | |
| Normal | 32 (69.6%) | 57 (65.5%) | 89 (66.9%) |
| Proteinuria and/or glycosuria* | 14 (30.4%) | 30 (34.5%) | 44 (33.1%) |
| Total recorded | 46 (100%) | 87 (100%) | 133 (100%) |

*Proteinuria—proteins in urine, glycosuria—sugar in urine. According to the NCD guideline, fasting blood glucose taken if overweight or have related symptoms, and urine analysis as indicated.

BP, blood pressure; COPD, chronic obstructive pulmonary disease; FBS, fasting blood sugar; NCD, non-communicable disease.

last appointment. Twenty-three (79%) patients perceived they were being treated by nurses with respect. Three patients commented on long waiting hours and lack of staff in facilities. Nearly all interviewed patients (n=25, 86%) agreed that nurses did a good job with explaining concepts and providing information on lifestyle advice and medication adherence in a clear and understandable manner. Satisfaction with the quality of care and treatment provision was also high (n=23, 79%). Most patients (n=24, 83%) expressed willingness to continue accessing facilities to receive NCD care; five patients, however, commented they would consider that only if

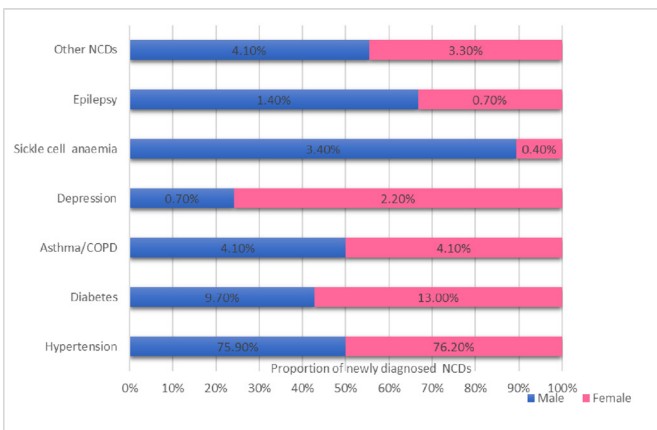

**Figure 1** Gender distribution of newly diagnosed non-communicable disease (NCD) cases. COPD, chronic obstructive pulmonary disease.

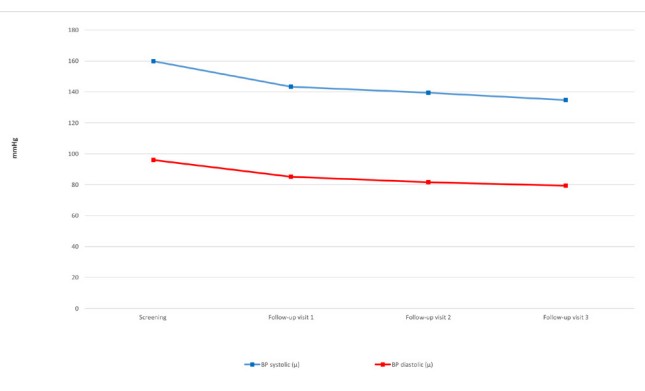

**Figure 2** Trends in blood pressure (BP) change among patients with hypertension (n=428).

the provision of care improves. When asked about challenges encountered during the period of clinic visits, 8 patients described no barriers, whereas two-thirds of the patients (n=19, 66%) described at least one barrier. Lack of money to buy the prescribed medication was the most frequently cited barrier (figure 3).

### Focus group discussions
#### Participants

FGs included 21 women and 2 men, between 31 and 60 years of age. All participants had post-secondary training in nursing. Some of the participants had prior experience of attending digital training. However, NCDs had not been prioritised in the health worker basic or post basic training the participants had previously received.

### Perceptions of mHealth training
#### Content

The mHealth training was stated to be a valuable experience. Participants were very appreciative of the knowledge they acquired about prevention and control of NCDs.

They talked about improved confidence and an increased sense of empowerment in their ability to provide counselling on lifestyle behaviours and medication adherence. This positively influenced their internal sense of importance, consistent with improved self-efficacy.

> Some of these cases … before now I didn't know how to handle them but now I am very confident in handling the issues. The manner in which I am now able to regulate clients' blood pressure is making me feel important. (FG1)

> I will begin by appreciating the NCD project and people. Before now, we actually didn't know how to handle these cases but now we are very confident in doing this. We know the particular drugs to prescribe and the dosages. (FG2)

#### Functionality

Participants perceived the functionality of mHealth technology and ease of use as important. Multiple benefits of online learning were noted including the ability to learn at your own pace, the opportunity to return to the

**Table 3** Trend on blood pressure from visit 1 to visit 3 based on the average of two measurements

| Parameter | Value | All patients (screening visit) | Patients with 2 visits | | patients with 3 visits | |
|---|---|---|---|---|---|---|
| | | | Visit 1 | Visit 2 | Visit 1 | Visit 3 |
| Systolic BP (mm Hg) | Valid values | 420 | 230 | 230 | 98 | 98 |
| | Mean (±SD) | 159.9 (23.3) | 160.3 (20.7) | 143.2 (20.6) | 162.1 (21.4) | 139.3 (19.1) |
| | Range | 90–270 | 90–220 | 99–218 | 90–220 | 80–180 |
| | Values>140 (%) | 77.6% | 79.6 | 43% | 83.7 | 32.3% |
| | Values>180 (%) | 15.0% | 14.8 | 3.9% | 16.3 | – |
| Diastolic BP (mm Hg) | Valid values | 420 | 230 | 230 | 98 | 98 |
| | Mean (±SD) | 96.0 (14.3) | 95.3 (13.5) | 85.5 (12.3) | 93.8 (15.3) | 81.7 (11.1) |
| | Range | 50–174 | 50–152 | 80–130 | 50–152 | 50–120 |
| | Values>90 (%) | 60.7% | 59.1 | 22.6% | 53.1 | 13.3% |
| | Values>110 (%) | 10.2% | 7.8 | 3.9% | 9.2 | 1.0% |

BP, blood pressure.

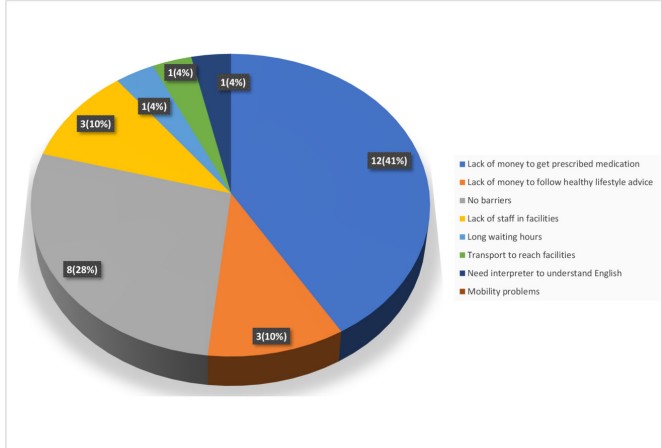

**Figure 3** Challenges encountered by patients during the period of clinic visits. *Count shows times each barrier was mentioned.

material when needed and using smart phones to access the NCD desk guide and modules.

> We are upgraded. It makes the provider more educated, because you read at your pace, you don't stress yourself, anytime you come back you read and understand. So… I think it was great. (FG1)

> It was a nice one. Almost all the trainings we have attended have not been this type. It has made us to know how to use our phone in learning online and also searching for other things you would not have ordinarily done because of the workload where we find ourselves. It is a good one. Many times, we tell ourselves let's go online, something is happening. (FG4)

### Experiences with the implementation of NCD care package
We interviewed 29 patients randomly selected across all the facilities. Of these, 21(72.4%) were satisfied with the main reason for dissatisfaction being our initiative's inability to provide medications.

#### Positive impact
Participants talked highly about the positive changes the integrated care delivery brought to the communities they served. Provision of BP and blood glucose diagnostic equipment meant targeted diagnostic and curative services delivery at point of access.

> The NCDP [the project] has made us have the necessary equipment to carry out blood pressure checks, and diagnosis of diabetes in my facility. This is a great achievement really. People come in now and say: "I want to check my blood pressure and sugar level." (FG4)

> When people come to our ANC, we have test kits to use and we have started testing. We even admitted some people. When people come to visit them, they go back to the community to tell others. Presently, we

have an increased client flow. It is helping us to detect these NCDs. (FG4)

The provision of PHC facility services dramatically improved relationships with communities and encouraged follow-up visits. During outreaches, health workers were able to use the donated sphygmomanometers to measure BP within the communities and refer people to the PHC facility.

> The relationship with the community as a result of the NCD is marvellous wherever we meet with these clients they greet one with respect and are open to us. (FG1)

> Providing the tests that were not existing in the community before has really helped to mobilize the people. They feel more relaxed and confident to come to the facility. (FG4)

Participants described a variety of community education strategies to raise awareness of the availability of NCD services. Some strategies involved providing monetary rewards for spreading the word about services by the town crier, and use of local worship places and/or community organisation to engage with communities.

> …I have a relationship with the town announcer. When people come and take the tests, we collect a token from them, and I use that to pay the town announcer. I type what I want him to announce. He reads through it and announces it to the people. From there, people started coming. (FG4)

> This project has impacted on our knowledge, and we can confidently do health talks and create awareness about blood pressure and other NCDs efficiently. My wish is…, I hope this project stays, honestly. (FG1)

### Challenges
Participants described challenges to the implementation of care packages as extending across clinical, organisational and societal levels.

#### Challenges at the clinical level
Challenges identified at the clinical level included increased workload associated with (1) ensuring the continuity of NCD care delivery, and (2) filling in the treatment cards. Participants stated that the success of theNCD initiative resulted in an increased flow of patients to the facilities for check-ups and follow-up consultations. Although this was generally perceived as a positive development, the increase in workload was highlighted.

> NCDs increased our work. Before now, we see them and they go away, we don't bother to follow them up or they don't come back to us to recheck their BP. Now we need to remind them to get their drugs and make sure their blood sugar levels are well controlled. For some of us, our facilities are easy to get to, so the clients can easily come there. Even when you are resting, they want you to come and check them. When

they heard that someone in the community died due to high blood pressure, that period a lot of people came for check-up . (FG4)

Some participants also noted that filling the treatment cards and registers was time-consuming and often discouraged their use. Content reduction was a proposed solution to address this.

Ordinarily we only have few questions to capture from clients but the NCD has increased the information that needs to be captured. Looking at this through other registers in the facility has made the workload high, and again there are series of tests that also need to be run on an individual basis. (FG1)

Technical glitches in the application and internet connection were also noted resulting in issues with uploading data into the central system. This resulted in double entry of data which was perceived as a double burden on staff. Some participants also recognised that supervision and monitoring provided by staff of the initiative helped them navigate many challenges along the way. .

### Challenges at the organisational level

The presence of professional support or opposition was highlighted as a key determinant of the successful integration of the care packages into daily routine. Participants talked at length about conflict of interest between the focal person responsible for NCD care and PHC facility in-charge. Prior to the training, there was no focal person responsible for NCD in these facilities. Following the NCD training, the 24 health workers were encouraged to assume the role of NCD focal person in their respective facilities. The PHC facility in-charges were typically registered nurses who also doubled as the administrative head of the health facilities. Examples of interference from the person-in-charge with modes of care delivery, equipment storage and use as well as generated income distribution were noted.

The thing is really disturbing because the in-charge kept telling you that the equipment is not for you and should be kept in the laboratory, giving unnecessary orders before you can make use of it. It pissed me off a lot. (FG1)

Some participants, on the other hand, talked positively about PHC facility in-charge and the support they received. They also acknowledged how some managers actively 'champion' the awareness-raising activities among communities, and how this considerably increased the flow of patients to facilities.

The new in charge has leadership skills. He went to the community heads and introduced himself, he went to churches. He always encouraged sick people to come and get tested. A lady died and it was discovered her blood pressure was high. So, they announced in all the churches that people should go and check their blood pressure and sugar level. This really helped us. (FG4)

Although many participants made reference to conflicts within the workplace, in general, negotiation, avoiding confrontations and/or training of managers on NCD care alongside junior staff were preferred approaches to resolving these problems.

I believe they should train them. Since we have this knowledge, they know some of the drugs, but they don't know the right time to administer. If they are also exposed to this training and they are told the stages and when to prescribe, they will know what to do. For example, in my facility my in-charge prescribed stage II drugs instead of stage I, I had to correct him. If they know, they will do better. (FG4)

Participants also noted negative changes in the spirit of collegiality and the need to ensure that 'the focal persons' have the expertise to manage NCD cases. This underscored the need for such training to be stepped down to other staff who may not have attended the initial face-to-face orientation meeting.

### Challenges at societal level

The restrictions to social interaction that followed the emergence of COVID-19 substantially limited the number of individuals attending the pilot PHCs for care. This had a negative impact on the number of persons who were eventually screened for NCDs. Also, some civil unrest occurred during the course of the project as part of the nationwide protests which resulted in vandalisation of some of the pilot PHCs. This halted services in the affected PHCs for several weeks and scaled back their recruitment rates after they reopened.

### Potential issues with the intervention scale-up

Participants talked about pressing issues in a healthcare system that potentially can have a negative impact on the scale-up of the intervention. The most commonly mentioned challenges were health personnel shortage (especially nurses), lack of knowledge on NCD management and control, lack of essential diagnostic equipment and medication necessary for effective NCD care delivery.

Potential solutions for ensuring the sustainability and potential scale-up of care packages at a national level were (1) targeted training of other staff on NCD prevention and management, and (2) task-shifting activities for NCD care. Participants also noted that staff at primary care level lacked basic knowledge on NCD care and would miss essential diagnostic tests if left to function at their current level. Therefore, they would be willing to provide training to junior staff to ensure effective delivery of NCD services.

Yes…, because some things, if left for the facility to carry on, will not be effective, but with a focal person he/she will ensure things are in order, since they know they will be held responsible. (FG1)

Yes, like in my facility I am the only staff. I will appreciate if another staff is sent here so that we can have more hands, and I will give the person on-the-job training or some kind of step-down training to enable whoever is coming to render the services. It will be bad if this NCDP [the NCD project] stops, before now we have had cases where somebody will just collapse while walking. The next thing you hear is people will be saying he is a ghost, not knowing is the effect of high blood pressure. So,…. I really don't want this NCD thing to just fade away like that, I don't know how to put it really. (FGD2)

## DISCUSSION
### Main findings
In this mixed methods pragmatic implementation of health worker training using an mHealth platform complemented by clinical supervision visits and a community of practice forum, we have demonstrated the feasibility of delivering training remotely leading to improvement in knowledge and skills in NCD screening, diagnosis and treatment. In addition, we have documented nurses' experiences, perspectives and the feasibility of task-shifting to nurses as well as patient satisfaction and challenges experienced with the care provided. This cohort of patients was found to have improved trends on average control of BP. There was a statistically significant reduction on both systolic and diastolic BP between the first and third visits among patients with hypertension. Nonetheless, the results do not prove a causal relationship between the mHealth training intervention and the resultant trend in BP. An experimental study with a control arm where health workers not involved in the study offered routine PHC level care at separate facilities would have provided necessary data for comparison. This was not possible in this study.

Diuretics and calcium channel blockers (nifedipine and amlodipine) were the most commonly prescribed antihypertensives, with metformin and glibenclamide being the most commonly prescribed therapies for diabetes mellitus. These prescribing practices were all in line with the recommendations within the NCD guide.

Health education/counselling was given to about two-thirds of patients on all visits. The process indicators demonstrate the ability of nurses in PHC clinics to successfully deliver the key components of NCD care. The NCD guide was designed for qualified staff at the level of a 3-year trained registered nurse. The deployment of an mHealth platform rather than a face-to-face classroom style model to deliver the training has been shown to be feasible and effective. This is seen in the improved knowledge scores on completion of the modules and assessment of skills seen in ability to initiate care for patients with NCDs.

The mHealth NCD training was perceived as a valuable experience with the health workers identifying multiple benefits of online learning and the integrated NCD care delivery package. This is important as there is not funding sufficient for face-to-face training (including per diems and venue cost) for nationwide scale-up of NCD training and supervision across Nigeria. As in other low-middle income countries there is no 'Global Fund' for NCDs in Nigeria, as there is for AIDS, TB and malaria. We provided a 1-day face-to-face orientation session to introduce the health workers to the mHealth platform and later supported them remotely to complete the modules online. This hybrid form of training appears to be suitable for the Nigerian context where face-to-face training of large numbers of health workers might not be feasible.

Challenges to the smooth rollout of this NCD package were the increased workload it generated, the technical problems with the internet connection and difficulty in securing the support of health centre in-charges for the project. Remote and periodic onsite supervision by local doctors and the provision of the materials (ie, BP equipment, glucometers) that were lacking at the facilities were additional strategies adopted to support this initiative. The availability of these materials along with the face-to-face orientation, provision of monthly funds for health workers' telecommunication fees and hard copies of the training materials contributed to the overall success of the pilot. Therefore, such extra provisions will need to be taken into consideration when scaling-up this NCD package. Considerations such as appropriate diagnosis, quality of care and linkage to continuing care will need to be further addressed using a health systems framework that deals with these challenges more comprehensively.[28]

While the pilot appears to be successful, the sustainability and replicability of this model of training and care initiation will depend on addressing several health system barriers such as appropriate staffing (not all facilities have a nurse), provision of medicines and diagnostic equipment, improving on the governance system at the facility level and formally task shifting to nurses the NCD care and prescribing. In addition, the reference to the initiative as a 'project' by health workers implies a risk to sustainability, if the NCD initiative is not wholly adopted by the state health management. Despite its implementation through the routine system by a local medical nongovernmental organisation (NGO), the perception of it as a 'project' may undermine the long-term sustainability at the facility level.

This mixed methods research adds to the evidence base of similar NCD interventions in resource-limited settings. Using a similar shared care guide for management of diabetes and hypertension care in Eswatini (southern Africa), nurses were able to successfully deliver diabetes and hypertension care with improvements in physiological parameters, although continued training and supervision were required to consolidate change.[14] A similar NCD guideline and training COMDIS-HSD partner initiative in Pakistan was found to be effective in trials and was scaled across the Punjab Province of 110 million population.[16–20] This Nigeria pilot adds evidence for the potential in a West African setting and for mHealth as an

alternative to expensive residential courses, which is a major constraint to NCD care scale-up.

## Limitations

The emergence of COVID-19 and imposed public health restrictions substantially limited the number of individuals attending the pilot PHCs to be screened and diagnosed over an equally short period of 10 months. Community mobilisation was conducted by the trained nurses to help overcome some social barriers to accessing care that came in the wake of the COVID-19 pandemic. Some of the health facilities were vandalised during the nationwide protests that engulfed Nigeria in October 2020, and this also had a negative impact on the NCD care services. In addition, the pragmatic nature of our initiative precluded the provision of medicines, which may have led to an increase in observed patients with NCD. Furthermore, given that many NGOs are operating in fragile health environments, pilot initiatives may not be subsequently implemented by ministries of health.

The observational nature of our study precludes us from drawing decisive conclusions as to whether care under the new NCD care model is better or worse than the alternative conventional residential training.

## CONCLUSION

This pilot initiative provides evidence of the feasibility of implementing an NCD care package supported by mHealth training. The complimentary support provided for health workers in the form of supervisory visits, a community of practice forum and ancillary equipment facilitated their ability to screen for NCDs in the PHCs. The improvement in BP measurements in patients at follow-up visits to the PHCs may be a reflection of the potential utility of this approach. These results support the implementation of such a package of care at scale within the Nigerian context and similar settings.

**Author affiliations**
[1]Department of Internal Medicine, College of Medical Sciences, University of Calabar, Calabar, Cross River State, Nigeria
[2]Foundation for Healthcare Innovation and Development, (FHIND), Calabar, Cross River State, Nigeria
[3]Department of Public Health, Faculty of Allied Medical Sciences, College of Medical Sciences, C, University of Calabar, Calabar, Cross River State, Nigeria
[4]Federal Ministry of Health, Abuja, Nigeria
[5]Queen Margaret University Institute for International Health and Development, Edinburgh, UK
[6]Instrat Global Health Solutions, Abuja, Nigeria
[7]Nuffield Centre for International Health and Development, Leeds Institute of Health Sciences, University of Leeds, Leeds, UK

**Acknowledgements** We appreciate the NCD division of the FMOH, Nigeria, Cross River State Ministry of Health and Primary Healthcare Development Agency as well as nurses and facility heads who participated in the pilot. We are grateful to all the patients who accepted to be managed using the NCD guides in the various health facilities in Cross River State Nigeria. We are thankful to Mr Tijesu Ojumu of the Nigeria Centre for Disease Control for designing the conceptual framework. Our sincere gratitude goes to the NIHR RUHF project of QMU Edinburgh for funding this initiative.

**Contributors** AAO, EEE, O Onwusaka and JW conceived the study. AAO, EEE, O Onwusaka, O Okuzu and CO were involved in the implementation of the programme. EEE, O Onwusaka, O Okuzu and CO handled the data entry. SA, AAO, O Onwusaka, JW and O Okuzu did the data cleaning and analysis. AAO, SA, EEE, JW and O Onwusaka wrote the first draft. All authors were involved in the reviewing and editing of this article and gave final approval of the version to be published. AAO is the guarantor of the article.

**Funding** This study/project is funded by the UK National Institute for Health Research (NIHR) (NIHR Global Health Research programme (project reference 16/136/100)/NIHR Research Unit on Health in Situations of Fragility).

**Competing interests** None declared.

**Patient and public involvement** Patients and/or the public were not involved in the design, or conduct, or reporting or dissemination plans of this research.

**Patient consent for publication** Not applicable.

**Ethics approval** The study received the approval of the Health Research and Ethics Committee of the Cross River State Ministry of Health Nigeria (Number/ID: CRSMOH/RP/REC/2019/173). Participants gave informed consent to participate in the study before taking part.

**Provenance and peer review** Not commissioned; externally peer reviewed.

**Data availability statement** Data are available upon reasonable request. Data generated by your research that supports your article will be made openly and publicly available on publication of your article.

**ORCID iDs**
Akaninyene Asuquo Otu http://orcid.org/0000-0002-6009-2707
Stella Arakelyan http://orcid.org/0000-0003-0326-707X

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
