## [Reviewer comments · BMJ Open]

ARTICLE DETAILS

TITLE (PROVISIONAL)	mHealth guideline training for non-communicable diseases in primary care facilities in Nigeria: a mixed methods pilot study
AUTHORS	Otu, Akaninyene; Effa, Emmanuel; Onwusaka, Obiageli; Omoyele, Chiamaka; Arakelyan, Stella; Okuzu, Okey; Walley, John

VERSION 1 – REVIEW

REVIEWER	Bogler, Lisa University of Göttingen, Centre for Modern Indian Studies
REVIEW RETURNED	10-Jan-2022

GENERAL COMMENTS	The authors describe a pilot study of a mobile health training on care for non-communicable diseases in primary health care facilities in Nigeria. The intervention is innovative and addresses a very relevant concern and it is commendable that the authors use both quantitative and qualitative methods to evaluate the intervention. My main concern is related to the choice of the primary outcome. In addition, some relevant information should be added to the manuscript to improve the understanding and evaluation of the intervention. Abstract: The description of the intervention should differentiate more clearly between actual intervention (including training via tablets) and evaluation of the intervention (including course evaluation, focus group discussions). Authors should provide clearer argumentation why improvement in blood pressure is used as primary outcome to evaluate the training. It does not seem to be possible to compare trends in blood pressure among patients where no intervention took place with trends in blood pressure among patients whose health workers received the training. This makes causal attribution difficult. Improvements in blood pressure could occur simply by repeated visits to the care facility if any treatment is given, irrespective of the training. Moreover, in the quantitative analysis, the authors seem to lump together patients who visited the clinic once, twice, and three times. For a trend analysis, it would be more convincing to look at patients who visited three times separately, as these could already have different levels of blood pressure at the first visit compared to those who visited only once. As of now, different samples are compared across the visits, making an interpretation of the trend difficult. In the discussion, the authors should at least mention that the analysis cannot identify the causal contribution of the intervention to the trend in blood pressure. The conclusion should therefore rather highlight other outcome measures as indication of feasibility and benefit of the training.
---

	More information should be provided regarding the following questions: How were PHCs and nurses selected for participation? Progress of module completion was tracked by the app. If possible, data on module completion should be presented. It would be interesting to have more information regarding the use of the WhatsApp group. How frequently was it used, what was discussed, was additional info on NCDs and treatment given by the clinical research team? More details should be provided about pre- and post-tests, including the maximum possible score, to allow for an interpretation of the reported scores. The described intervention was much more intensive than a provision of access to the mobile health training. It also seems to have included a face-to-face orientation, material, equipment, WhatsApp group, weekly calls, and facility visits. This should be critically discussed when considering feasibility for scale-up. Results: Reporting in second paragraph is not clear. 561 community members were diagnosed with NCD and 569 were served. Does that mean only 8 patients did not have any NCD? If so, what do the 71% refer to? "Average age of NCD diagnosis for males" is an unclear phrase. Minor comments:  - Abbreviations should be spelled out at first use (e.g. NCD and PHC in the abstract). - Introduction, page 3, line 17: "The rapid global epidemiologic transition and rising rates of NCDs ARE also having" instead of "IS". - Introduction, page 3, line 34: "3.3% of the budget is DEvoted to mental health care..." instead of "voted to". - Focus group discussions, page 6, line 26: 24 participants are mentioned while previously, it says 23 participants. Please clarify. - Quantitative data, page 7, line 17: "assess the effectiveness" instead of "access" - Healthy lifestyle advice, page 10: If possible, please add numbers here as well. In the discussion, it says "the large majority of patients" received such advice but this should be backed up by numbers. - Experiences with the implementation of NCD care package, page 12, line 53: "refer people to" instead of "refer in people to". - Challenges, page 15, line 9: In the introduction (page 4, line 25), it says that certificates were part of the intervention, while here the sentence suggests that no certificates were given for the training. Please clarify.
--	---

REVIEWER	Zimba, Chifundo UNC Project-Malawi
REVIEW RETURNED	27-Jan-2022

GENERAL COMMENTS	 1. Title: Include the approach used ie mixed method 2. Abstract should be structured using the journal structure. Also include background information; Your purpose is not clear; on methods you have only focused on design 3. Introduction: Include hypothesis for your quantitative component 4. Method section needs to be reorganized. Describe in detail how you selected your sample, how you collected data both in
--

	qualitative and quantitative, include description of data collection instruments; data analysis for qualitative needs to be detailed. No triangulation of qual and quant data 5. Ethical consideration not stated anywhere in the paper 6. Results: No data collected via direct observation stated in the paper; some qualitative data stated as "challenges" but the narratives are "positives"; no section indication results after triangulation; no "fidelity" results on how nurses were recording on the cards after training over time 7. Discussion needed more work. Some results discussed are not stated on the results section; include how your results contribute to science i.e. what is different apart from the similarities that you have stated. 8. General: There are some narrative disjoints in some sections eg covid information under focus group discussion and on FGD results participants and perceptions of m-health training...I cant understand the gap
--	--

VERSION 1 – AUTHOR RESPONSE

Comment	Response
Abstract: The description of the intervention should differentiate more clearly between actual intervention (including training via tablets) and evaluation of the intervention (including course evaluation, focus group discussions).	Accepted. We have rewritten the abstract and provided a clearer differentiation of the intervention and evaluation components.
Authors should provide clearer argumentation why improvement in blood pressure is used as primary outcome to evaluate the training. It does not seem to be possible to compare trends in blood pressure among patients where no intervention took place with trends in blood pressure among patients whose health workers received the training. This makes causal attribution difficult. Improvements in blood pressure could occur simply by repeated visits to the care facility if any treatment is given, irrespective of the training. Moreover, in the quantitative analysis, the authors seem to lump together patients who visited the clinic once, twice, and three times. For a trend analysis, it would be more convincing to look at patients who visited three times separately, as these could already have different levels of blood pressure at the first visit compared to those who visited only once. As of now, different samples are compared across the visits, making an interpretation of the trend	Points taken. Our choice of hypertension as the primary outcome is because it is prevalent and easy to record compared to other NCDs such as Diabetes Mellitus where assessment of glycaemic control with glycated haemoglobin would have required a longer follow up time than afforded by this pilot. We agree that we are unable to compare trends of blood pressure control since this was a pilot and we focused on the feasibility and scalability of the intervention. We have not compared the outcome with facilities where no training took place. We have reflected our inability to show causality between the mhealth training and blood pressure trend in the discussion section.

Comment	Response
difficult. In the discussion, the authors should at least mention that the analysis cannot identify the causal contribution of the intervention to the trend in blood pressure. The conclusion should therefore rather highlight other outcome measures as indication of feasibility and benefit of the training.	
More information should be provided regarding the following questions: How were PHCs and nurses selected for participation? Progress of module completion was tracked by the app. If possible, data on module completion should be presented. It would be interesting to have more information regarding the use of the WhatsApp group. How frequently was it used, what was discussed, was additional info on NCDs and treatment given by the clinical research team?	Purposive sampling was used to choose the PHCs. It was spread across the various senatorial districts and local government areas. The 10 PHCs were meant to reflect 10% of the 190 large PHCs in Cross River State We have reflected this in the section 'Setting and participants' The completion rate was 100%. The app was designed such that participants mandatorily had to complete all modules. The threshold score for completion of each module was 70 % The WhatsApp group served as a platform for real time trouble shooting of clinical and technical issues supported by the clinicians and IT staff on the research team. We have reflected this in the methods section.
More details should be provided about pre- and post-tests, including the maximum possible score, to allow for an interpretation of the reported scores.	The maximum possible score was 100% Pass score was set at 70%. We now have included this in the manuscript
The described intervention was much more intensive than a provision of access to the mobile health training. It also seems to have included a face-to-face orientation, material, equipment, WhatsApp group, weekly calls, and facility visits. This should be critically discussed when considering feasibility for	Accepted. As above, RE the WhatsApp group in the methods section. Further, in response to your points we have added a logic model table with the intervention components. We have also expanded our discussion on the implications of the supportive measures to the mhealth training on feasibility for scale up

Comment	Response
scale-up.	
Results: Reporting in second paragraph is not clear. 561 community members were diagnosed with NCD and 569 were served. Does that mean only 8 patients did not have any NCD? If so, what do the 71% refer to? “Average age of NCD diagnosis for males” is an unclear phrase.	We have looked at our data again and corrected this error. 414 (72.7%) out of the 569 patients seen were newly diagnosed cases of NCDs. The rest were patients who had previously been diagnosed with NCDs prior to the onset of project. This has been rephrased and now reads ‘Average age of males diagnosed with an NCD’
Minor comments:  - Abbreviations should be spelled out at first use (e.g. NCD and PHC in the abstract). - Introduction, page 3, line 17: “The rapid global epidemiologic transition and rising rates of NCDs ARE also having” instead of “IS”. - Introduction, page 3, line 34: “3.3% of the budget is devoted to mental health care...” instead of “voted to”. - Focus group discussions, page 6, line 26: 24 participants are mentioned while previously, it says 23 participants. Please clarify. - Quantitative data, page 7, line 17: “assess the effectiveness” instead of “access” - Healthy lifestyle advice, page 10: If possible, please add numbers here as well. In the discussion, it says “the large majority of patients” received such advice but this should be backed up by numbers. - Experiences with the implementation of NCD care package, page 12, line 53: “refer 	We have corrected this This has been corrected This has been corrected While 24 were invited, 23 eventually participated This has been corrected 356 (62.6 %) patients were given lifestyle advice such as reducing/ stopping cigarette smoking and alcohol, reducing weight, reducing salt intake, increasing physical activity/ exercise, reducing stress, medication adherence, healthy eating, and healthy living. We have reworded this section and added numbers as suggested. This has been corrected

Comment	Response
people to" instead of "refer in people to". -	
Challenges, page 15, line 9: In the introduction (page 4, line 25), it says that certificates were part of the intervention, while here the sentence suggests that no certificates were given for the training. Please clarify	Certificates were issued to the participants who completed the NCD modules. We have clarified this in the manuscript
Title: Include the approach used ie mixed method	Thank you for this suggestion. We have included 'mixed methods' in the title
2. Abstract should be structured using the journal structure.	Thank you for this comment. We have restructured the abstract for clarity.
Also include background information; Your purpose is not clear; on methods you have only focused on design	We have included this information in the 'objectives' section of the abstract.
3. Introduction: Include hypothesis for your quantitative component	We have updated the introduction as suggested with the text below: We hypothesised that mHealth NCD training of nurses should lead to improved knowledge and improved competencies in the screening/diagnosis, treatment initiation and optimal blood pressure control in patients at PHC level.
4. Method section needs to be reorganized. Describe in detail how you selected your sample, how you collected data both in qualitative and quantitative, include description of data collection instruments; data analysis for qualitative needs to be detailed. No triangulation of qual and quant data	We have provided further details of our sampling strategy. Details of data collection for both quantitative and qualitative components of the study have already been provided in the manuscript. Although we have used a mixed methods approach, each component focused on a different outcome of the study and so we are unable to triangulate the data.
5. Ethical consideration not stated anywhere in the paper	The study was approved by the Cross River State Health Research Ethics Committee We have now included this statement in the updated manuscript
6. Results: No data collected via direct observation stated in the paper; some qualitative data stated as "challenges" but the narratives are "positives"; no section indication results after triangulation; no "fidelity" results on how nurses were recording on the cards after training over time	Although we observed nurses attend to patients during visits and studied entries in the cards, this was in the context of clinical supervision and not necessarily evaluation. Therefore, we did not collect data by observation and triangulation was not performed.

Comment	Response
	We have noted the “positives” in the “challenges” section. These have been included to highlight perceived solutions especially in the section “challenges at the organisational level”
7. Discussion needed more work. Some results discussed are not stated on the results section; include how your results contribute to science i.e. what is different apart from the similarities that you have stated.	We have updated the discussion section
8. General: There are some narrative disjoints in some sections eg covid information under focus group discussion and on FGD results participants and perceptions of m-health training...I cant understand the gap	Our intention was to highlight the effect of covid on the subsequent recruitment of participants for the study. For clarity in the narration, we have moved the section on covid to the discussion on limitations.

VERSION 2 – REVIEW

REVIEWER	Bogler, Lisa University of Göttingen, Centre for Modern Indian Studies
REVIEW RETURNED	13-Apr-2022

GENERAL COMMENTS	The authors addressed several of my comments in this revised version of the study. However, some concerns remain unaddressed and a few new points arose through the revision. Most importantly, I still believe that 1) the approach for the quantitative assessment of trends in blood pressure over time is not appropriate and 2) the discussion does not reflect the fact that the intervention was more than the provision of the mhealth platform, making scale-up more difficult. I hope that these points can be addressed because the project is very interesting and could be a valuable approach for improving NCD care in low-resource settings. Comments in order of appearance: Methods Implementation package: The authors replaced figure 1 with table 1. While this table supposedly clarifies the intervention input and project design, there are several issues related to this table. The authors should explain in the text what the purpose of the table is, instead of listing what items it contains. Especially the sentence “Included in brackets were the research pre/post training score, ...” is not informative. What does it show – the theory of change and how it was evaluated? If the brackets are meant to contain the items related to project evaluation, this is not made clear and is not designed optimally. Additional small issues:  - If the table is supposed to list all intervention inputs, it misses the monthly funds for health workers’ telecommunication fees. - The asterisk in column 2 does not seem to be explained anywhere.
---

	- The bullet point in column 1 “Clinical and implementation, nurse WhatsApp support group” is not clear. Setting and participants: The authors added the explanation of how PHCs were chosen. How nurses were selected remains an open question and should be answered in this section. Does each PHC have only one nurse (or two) nurses and all nurses of each PHC were selected? If this is the case, it should simply be added. Results In the methods section (Process), the authors state that 70% was the pass score for each module. It would be helpful if the authors added at the beginning of the results section whether all participating nurses actually achieved this pass score or what the score was. It would be very informative to know to what extent the nurses were able to complete the modules in the given two weeks. The numbers in the second paragraph of the results section have been corrected. Still, the explanation of newly diagnosed and previously diagnosed cases could be clarified in the text simply by using the same description as in the response to previous comment on this. Primary outcome: The numbers reported in the text do not equal the numbers in the table for systolic and diastolic blood pressure at visit 3. The authors still lump together all patients in the analysis of the trend in blood pressure. This is one of the main concerns. Patients who visit only once, who visit twice, who visit three times, might be different from each other. This makes a comparison between blood pressure at visit 1 and visit 3 very difficult because the sample for which the mean is calculated at visit 1 is different from the sample for which the mean is calculated at visit 3. The authors should look at the groups separately. Only for patients who visited three times, one could compare the blood pressure at their first and their third visit. For patients who visited twice, one could compare the blood pressure at their first and second visit. If the authors have concerns about splitting the sample for these comparisons, they should at least show that the three groups are not different from each other. At a minimum, they should report the mean blood pressure for each group at the first visit. While other characteristics could still be different between the three groups, that would at least show whether their blood pressure was similar at the first visit. If it is not, the decreasing trend could simply be driven by the group visiting three times having had a lower blood pressure to begin with. This would invalidate the argument that blood pressure improved over time. Challenges at the organizational level: The authors report conflicts between the “focal person responsible for NCD care” and the “PHC facility in-charge”. Are the trained nurses the “focal person responsible for NCD care”? Who is the PHC facility in-charge? Is this typically a doctor or maybe an administrative person? This should be clarified. Potential issues with intervention scale-up: The last sentence of the first paragraph says that “The intervention included to train one nurse”. Does this mean one nurse per PHC? This also relates
--	--

	back to the unclear selection of nurses. What is the purpose of this sentence? Discussion Main finding The authors state that the study has documented the satisfaction of clients, but this seems to be only reported through the experience of nurses, not the clients themselves. This should be rephrased. The authors state that the “mhealth platform rather than face-to-face classroom style model to deliver the training” was feasible. However, the platform was also introduced with a face-to-face meeting and training that included group discussions and practical sessions. Moreover, all nurses reportedly received hard copies of the guidelines, equipment was provided. These components of the intervention should not be overlooked here and this relates to my second main concern. The platform might not be used as much without the face-to-face training or required equipment for testing. There is no comment on the use of the hard copy instead of the online modules. One argument made by the authors is that there is no funding for the face-to-face training. So how should this be scaled up? This is still not sufficiently reflected in the discussion. Reducing the intervention to the mhealth platform in a scale-up might severely diminish its reported success. Reducing the intervention to the mhealth platform in its description biases the conclusion regarding the feasibility of scale-up. Some notes regarding typos, punctuation:  - The authors should read through manuscript without tracked changes. The tracked changes lead to many details being overlooked. Some examples are listed here. - Abstract: Setting: “NCDs was not part ...” should be „NCDs were not part” - Add space between words  o Table 1:Logic o Background, last paragraph: “Included, in bracketswere ...” should be “Included in brackets were ...” o Page 18, last paragraph: “contributed tothe overall success” o Page 19: “sustainability at facility level.This mixed methods research” - Page 5, first paragraph: “there is not funding” should be “there is no funding”. - Setting and participants: In the middle of the paragraph, there is a standalone “PHCs”. This should be removed. - Page 5, second paragraph: “training focused hypertension and/or diabetes” should be “training focused on ...” and is it “and” or “or”? - Page 8, second paragraph: “troubleshooting upload of data” - This needs some preposition such as maybe “troubleshooting in upload of data”. - Page 8, second paragraph: “The WhatsApp group also served ...” ends with two full stops. - Page 8, last line: “FDG” should be “FGD” - Focus group discussions: “Shorts message services invitation” should be “Short message services invitations” - Healthy lifestyle advice: First sentence in this paragraph ends with two full stops. - Page 15: “FD4” should be “FG4” - Page 16: “FD1” should be “FG1” - Page 18, second paragraph: Paragraph ends with closing bracket that is not opened.
--	--

	 - Page 18, last paragraph: “difficultly” should be “difficulty”. - Page 18, last paragraph: “availability of these se materials” should be “availability of these materials” - Throughout the manuscript: Why “guide(line)” and not either “guide” or “guideline”? The reader should not have to decide what it actually is.
--	--

REVIEWER	Zimba, Chifundo UNC Project-Malawi
REVIEW RETURNED	23-May-2022

GENERAL COMMENTS	The paper has been well revised. Few this as indicated on the paper to be reviewed
--

VERSION 2 – AUTHOR RESPONSE

Response to reviewer comments

Reviewer comment	Response
- Please include a copy of the focus group guide that was used to collect data in this study (ie, Open-ended questions used during the focus group discussions).	We have now provided a copy of the focus group guide as an appendix
- Please note that p-values cannot be equal to 0. Please amend as necessary.	We have amended the p-values accordingly
1) the approach for the quantitative assessment of trends in blood pressure over time is not appropriate	We feel we have adopted an acceptable approach to the assessment of trends in blood pressure
2) the discussion does not reflect the fact that the intervention was more than the provision of the mhealth platform, making scale-up more difficult.	We have reflected the other aspects of the intervention which contributed to its success in the discussion i.e. the complementary support provided for health workers in the form of supervisory visits, a community of practice forum and ancillary equipment facilitated their ability to screen for NCDs in the PHCs. We have stated that scale up will need to address all these issues to be successful
Implementation package: The authors replaced figure 1 with table 1. While this table supposedly clarifies the intervention input and project design, there are several issues related to this table. The authors should explain in the text what the purpose of the table is, instead of listing what items it contains. Especially the sentence “Included in brackets were the	We have now provided a detailed explanation of the logic model table within the manuscript. We have also deleted the sentence which was not informative

Reviewer comment	Response
research pre/post training score, ..." is not informative. What does it show – the theory of change and how it was evaluated? If the brackets are meant to contain the items related to project evaluation, this is not made clear and is not designed optimally.	
Additional small issues:  - If the table is supposed to list all intervention inputs, it misses the monthly funds for health workers' telecommunication fees. 	The monthly funds for health workers' telecommunication fees have now been included as an input in Table 1
 - The asterisk in column 2 does not seem to be explained anywhere. 	We have deleted the asterisk
 - The bullet point in column 1 "Clinical and implementation, nurse WhatsApp support group" is not clear. 	We have edited this to: Nurse WhatsApp support group
Setting and participants: The authors added the explanation of how PHCs were chosen. How nurses were selected remains an open question and should be answered in this section. Does each PHC have only one nurse (or two) nurses and all nurses of each PHC were selected? If this is the case, it should simply be added	The 23 nurses were proposed for the training by the Cross River State Primary Health Care Development Agency (CRSPHCDA). We have now provided this information in the manuscript
Results In the methods section (Process), the authors state that 70% was the pass score for each module. It would be helpful if the authors added at the beginning of the results section whether all participating nurses actually achieved this pass score or what the score was. It would be very informative to know to what extent the nurses were able to complete the modules in the given two weeks.	We have provided details on the number of nurses who achieved the pass score of 70% in the tests and to what extent the nurses were able to complete the modules in the given two weeks.
The numbers in the second paragraph of the results section have been corrected. Still, the explanation of newly diagnosed and previously	This has been done

Reviewer comment	Response
diagnosed cases could be clarified in the text simply by using the same description as in the response to previous comment on this.	
Primary outcome: The numbers reported in the text do not equal the numbers in the table for systolic and diastolic blood pressure at visit 3.	We have corrected this
The authors still lump together all patients in the analysis of the trend in blood pressure. This is one of the main concerns. Patients who visit only once, who visit twice, who visit three times, might be different from each other. This makes a comparison between blood pressure at visit 1 and visit 3 very difficult because the sample for which the mean is calculated at visit 1 is different from the sample for which the mean is calculated at visit 3. The authors should look at the groups separately. Only for patients who visited three times, one could compare the blood pressure at their first and their third visit. For patients who visited twice, one could compare the blood pressure at their first and second visit. If the authors have concerns about splitting the sample for these comparisons, they should at least show that the three groups are not different from each other. At a minimum, they should report the mean blood pressure for each group at the first visit. While other characteristics could still be different between the three groups, that would at least show whether their blood pressure was similar at the first visit. If it is not, the decreasing trend could simply be driven by the group visiting three times having had a lower blood pressure to begin with. This would invalidate the argument that blood pressure improved over time.	We reflected patients who attended all three clinic visits i.e. the patients who attended the third visit also attended the second and first visits. Therefore, we feel it is accurate to state that the blood pressure improved with time > This information is already reflected in figure 2. We have however, relabeled table 3 to show a screening visit and added a column to reflect the follow up visit 3. We have re-run the analysis as requested. We have now separated the hypertensive patients into 3 groups. Group 1(screening for all)- everyone at their first visit Group 2 – only those with at least 2 visits Group 3 – only those with at least 3 visits. For group 2, we have compared their systolic and diastolic BP at visits 1 and 2. For group 3 , we have compared their BP at visits 1 and 3. We have updated the results in the tables.
Challenges at the organizational level: The authors report conflicts between the “focal person responsible for NCD care” and the “PHC facility in-charge”. Are the trained nurses the “focal person responsible for NCD care”? Who is the PHC facility in-charge? Is this typically a doctor or maybe an administrative person? This should be clarified.	Prior to the training, there was no “focal person” responsible for NCD in these facilities. Following the sensitization and training, the nurses in question assumed this role. The PHC facility in-charge is typically a nurse who also doubles as the administrative head. We have now included these details in the manuscript

Reviewer comment	Response
Potential issues with intervention scale-up: The last sentence of the first paragraph says that “The intervention included to train one nurse”. Does this mean one nurse per PHC? This also relates back to the unclear selection of nurses. What is the purpose of this sentence?	For clarity, we have deleted this sentence. We have added a sentence indicating that on the average there is one nurse per facility in the ‘Setting and participants’ section
Discussion Main finding The authors state that the study has documented the satisfaction of clients, but this seems to be only reported through the experience of nurses, not the clients themselves. This should be rephrased.	We interviewed 29 patients randomly selected across all the facilities. Of these, 21(72.4%) were satisfied with the main reason for dissatisfaction being our initiative’s inability to provide medications.
The authors state that the “mhealth platform rather than face-to-face classroom style model to deliver the training” was feasible. However, the platform was also introduced with a face-to-face meeting and training that included group discussions and practical sessions. Moreover, all nurses reportedly received hard copies of the guidelines, equipment was provided. These components of the intervention should not be overlooked here and this relates to my second main concern. The platform might not be used as much without the face-to-face training or required equipment for testing. There is no comment on the use of the hard copy instead of the online modules.	The mHealth platform was introduced with a face-to-face orientation session and supported by hard copies of the training materials. We have reflected these additional factors in the discussion and stated that these will need to be factored in during a scale up of the training package.
One argument made by the authors is that there is no funding for the face-to-face training. So how should this be scaled up? This is still not sufficiently reflected in the discussion. Reducing the intervention to the mhealth platform in a scale-up might severely diminish its reported success. Reducing the intervention to the mhealth platform in its description biases the conclusion regarding the feasibility of scale-up.	The traditional face-to-face training which can take several days is not feasible in Nigeria. However, a hybrid model that consists of a brief face-to-face orientation on the mHealth platform followed by use of the online materials with support from experienced people is the model we are proposing. We have stated this clearly in the discussion
Some notes regarding typos, punctuation:  - The authors should read through manuscript without tracked changes. The tracked changes 	We have comprehensively addressed all the typos and punctuation errors

Reviewer comment	Response
lead to many details being overlooked. Some examples are listed here.  - Abstract: Setting: “NCDs was not part ...” should be „NCDs were not part” - Add space between words o Table 1:Logic o Background, last paragraph: “Included, in brackets were ...” should be “Included in brackets were ...” o Page 18, last paragraph: “contributed tothe overall success” o Page 19: “sustainability at facility level.This mixed methods research” - Page 5, first paragraph: “there is not funding” should be “there is no funding”. - Setting and participants: In the middle of the paragraph, there is a standalone “PHCs”. This should be removed. - Page 5, second paragraph: “training focused hypertension and/or diabetes” should be “training focused on ...” and is it “and” or “or”? - Page 8, second paragraph: “troubleshooting upload of data” - This needs some preposition such as maybe “troubleshooting in upload of data”. - Page 8, second paragraph: “The WhatsApp group also served ...” ends with two full stops. - Page 8, last line: “FDG” should be “FGD” - Focus group discussions: “Shorts message services invitation” should be “Short message services invitations” - Healthy lifestyle advice: First sentence in this paragraph ends with two full stops. - Page 15: “FD4” should be “FG4” - Page 16: “FD1” should be “FG1” - Page 18, second paragraph: Paragraph ends with closing bracket that is not opened. 	

Reviewer comment	Response
 - Page 18, last paragraph: “difficultly” should be “difficulty”. - Page 18, last paragraph: “availability of these se materials” should be “availability of these materials” - Throughout the manuscript: Why “guide(line)” and not either “guide” or “guideline”? The reader should not have to decide what it actually is 	
Which ones are strengths and which ones are limitations?	The first two bullet points refer to strengths while the last two reflect the limitations
You mentioned about sickle cell above. Where is its training package?	The NCD training package covered several diseases of which sickle cell was just one of them. There was not a separate training module for sickle cell
Say more about this. Ie what is it and what factors you followed to come up with your sample? You may include “how”?	We have provided more details on the Cross River State Primary Health Care Development Agency (CRSPHCDA). We have also given more information on how we came up with our sample
Your sample was “nurses”. Are these nurses or another cadre	The clinicians we are referring to were members of the research team who provided support to the trained nurses. These clinicians were not staff of the CRS government. We have provided some clarification of this point in the manuscript
But you may cite some papers to show whether your sample is supported by literature or not. Why did you come up with 24 people?	Our sample was not supported by literature. This was an implementation pilot and we simply trained the nurses who were proposed by the CRSPHCDA. The numbers we trained is a reflection of the number of registered nurses available in these health facilities. On the average, there is one nurse per PHC facility.
So what?	We are appreciative of the value of training focal persons to ensure they have the expertise to manage NCD cases. This point is reflected in the manuscript. We have stated: “This point reflected the need for such training to be stepped down to other staff who may not have

Reviewer comment	Response
	attended the initial face-to-face orientation meeting”
Limitations and strength stated at the beginning of the paper are somehow different from this section Also how covid-19 affected the study is not clearly stated within the paper	We have edited the flow of the article to align with the strengths and limitations we stated earlier on in the article. We have also reflected how covid1-9 affected the study within the article
This statement can be splinted into 2	We have splinted the conclusion in two